# Multi-dimensional genomic analysis of myoepithelial carcinoma identifies prevalent oncogenic gene fusions

Martin G. Dalin[1,2,3], Nora Katabi[4], Marta Persson[5], Ken-Wing Lee[1], Vladimir Makarov[1,6], Alexis Desrichard [1,6], Logan A. Walsh[1], Lyndsay West[7], Zaineb Nadeem[1,7], Deepa Ramaswami[1,7], Jonathan J. Havel[1,6], Fengshen Kuo [1,6], Kalyani Chadalavada[8], Gouri J. Nanjangud [8], Ian Ganly[7], Nadeem Riaz[6,9], Alan L. Ho[10], Cristina R. Antonescu[4], Ronald Ghossein[4], Göran Stenman [5], Timothy A. Chan[1,6,9] & Luc G.T. Morris [1,6,7]

Myoepithelial carcinoma (MECA) is an aggressive salivary gland cancer with largely unknown genetic features. Here we comprehensively analyze molecular alterations in 40 MECAs using integrated genomic analyses. We identify a low mutational load, and high prevalence (70%) of oncogenic gene fusions. Most fusions involve the *PLAG1* oncogene, which is associated with *PLAG1* overexpression. We find *FGFR1-PLAG1* in seven (18%) cases, and the novel *TGFBR3-PLAG1* fusion in six (15%) cases. *TGFBR3-PLAG1* promotes a tumorigenic phenotype in vitro, and is absent in 723 other salivary gland tumors. Other novel *PLAG1* fusions include *ND4-PLAG1*; a fusion between mitochondrial and nuclear DNA. We also identify higher number of copy number alterations as a risk factor for recurrence, independent of tumor stage at diagnosis. Our findings indicate that MECA is a fusion-driven disease, nominate *TGFBR3-PLAG1* as a hallmark of MECA, and provide a framework for future diagnostic and therapeutic research in this lethal cancer.

[1] Human Oncology and Pathogenesis Program, Memorial Sloan Kettering Cancer Center, New York, NY 10065, USA. [2] Department of Pediatrics, Institute of Clinical Sciences, University of Gothenburg, 41685 Gothenburg, Sweden. [3] Queen Silvia Children's Hospital, Sahlgrenska University Hospital, 41685 Gothenburg, Sweden. [4] Department of Pathology, Memorial Sloan Kettering Cancer Center, New York, NY 10065, USA. [5] Sahlgrenska Cancer Center, Department of Pathology and Genetics, University of Gothenburg, 40530 Gothenburg, Sweden. [6] Immunogenomics and Precision Oncology Platform, Memorial Sloan Kettering Cancer Center, New York, NY 10065, USA. [7] Head and Neck Service, Department of Surgery, Memorial Sloan Kettering Cancer Center, New York, NY 10065, USA. [8] Molecular Cytogenetics Core Facility, Memorial Sloan Kettering Cancer Center, New York, NY 10065, USA. [9] Department of Radiation Oncology, Memorial Sloan Kettering Cancer Center, New York, NY 10065, USA. [10] Head and Neck Medical Oncology Service, Department of Medicine, Memorial Sloan Kettering Cancer Center, New York, NY 10065, USA. Martin G. Dalin and Nora Katabi contributed equally to this work. Correspondence and requests for materials should be addressed to T.A.C. (email: chant@mskcc.org) or to L.G.T.M. (email: morrisl@mskcc.org)

Myoepithelial carcinoma (MECA) is an understudied type of salivary carcinoma, characterized by aggressive clinical behavior and a high rate of distant metastases. There are no active systemic therapies for this cancer, rendering recurrent or metastatic disease generally incurable. The molecular alterations that define MECA have not been well characterized[1, 2]. In earlier studies, MECA constituted less than 2% of all salivary gland cancers, but its incidence is now believed to be higher due to increased recognition by pathologists[3]. In fact, MECA is the second most common type of carcinoma arising from transformed benign salivary adenomas[4]. MECAs most commonly arise in the parotid gland, followed by the submandibular and minor salivary glands[1]. While most cases present as localized disease, advanced locoregional or distant recurrence is relatively common, leading to a 5-year overall survival rate of 64%[2].

In around half of the cases, MECA is the result of a benign pleomorphic adenoma (PA) undergoing malignant transformation (denoted MECA ex-PA)[1, 4]. Microscopically, MECAs have heterogeneous morphologic and immunohistochemical features. The presence of carcinoma ex-PA, as well as histological findings of tumor necrosis and vascular invasion, are negative prognostic factors in MECA[5].

Rearrangements affecting the pleomorphic adenoma gene 1 (PLAG1) or high-mobility group AT-hook 2 gene (HMGA2) have been observed in PAs and MECA ex-PAs[6–11]. However, PLAG1 rearrangements have not been detected in MECA de novo, and the repertoire of PLAG1 fusion partners in MECA ex-PA remains poorly defined. Apart from rearrangement of PLAG1 and HMGA2, the spectrum of genetic alterations in MECA is unknown.

In order to guide clinical investigation for understudied, lethal cancers such as MECA, understanding the genetic alterations that underlie this disease is critical. Identifying molecular alterations may nominate targetable signaling pathways, and guide further research with targeted or immunotherapeutic approaches. In this study, we performed multi-dimensional analyses on 40 MECAs, including whole-exome and RNA sequencing, array comparative genomic hybridization, fluorescence in situ hybridization, and protein expression analyses. Taken together, our results provide the first molecular characterization of MECA, providing novel information on the molecular underpinnings of this cancer, which may guide future clinical investigation in this aggressive disease.

## Results

**Mutation spectrum of MECA.** To investigate the mutational landscape of MECA, we initially performed whole-exome sequencing of DNA from snap-frozen tumor and matched normal tissue of 12 patients (cohort 1). The mean coverage was

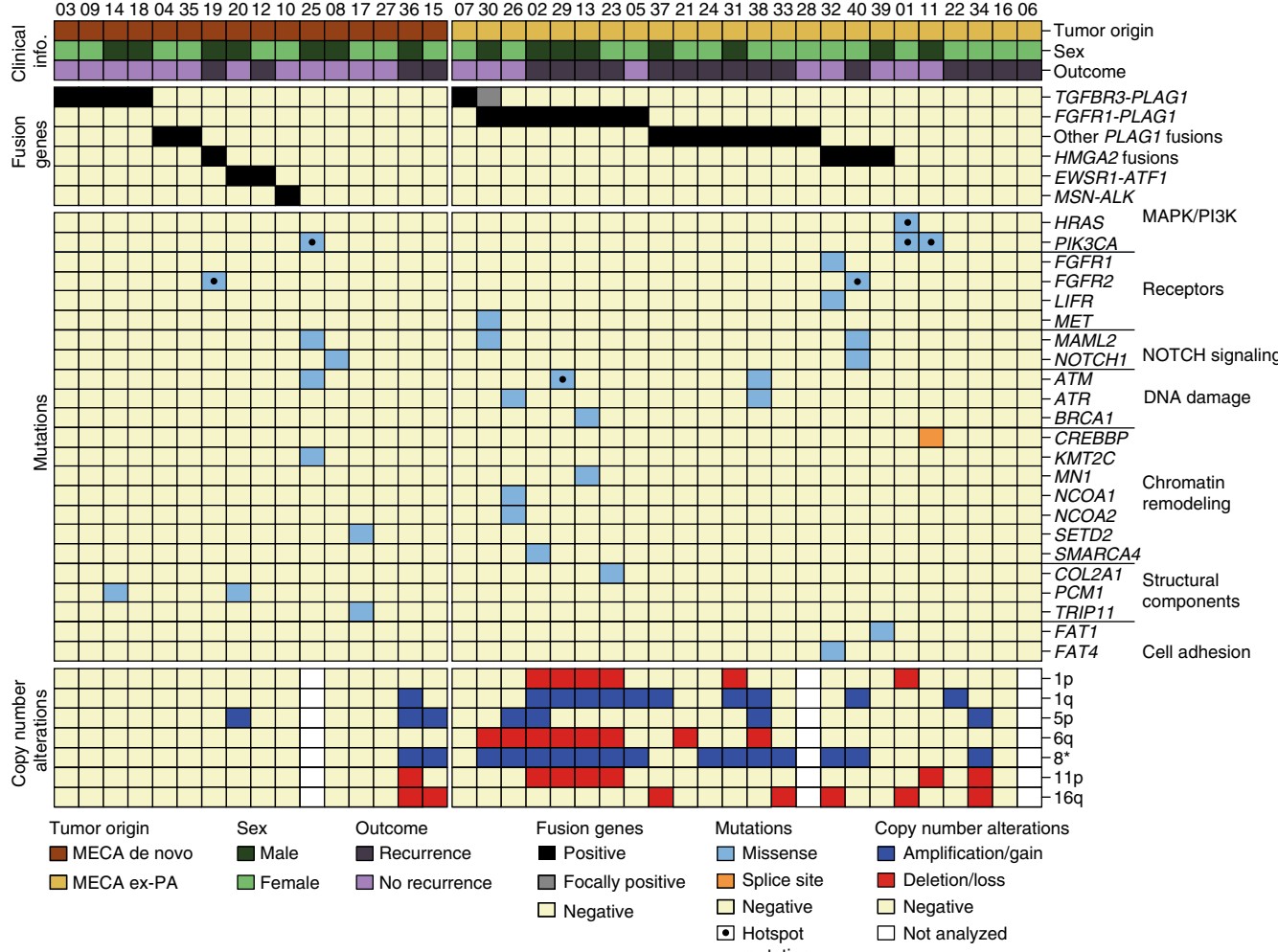

**Fig. 1** Genetic landscape of MECA. Clinical information, fusion genes, mutations, and arm-level copy number alterations in 40 cases of MECA. Mutations of genes listed in cancer census that are reported in more than one case in the COSMIC database are shown. Hotspot mutations are those reported in more than 25 cases in COSMIC

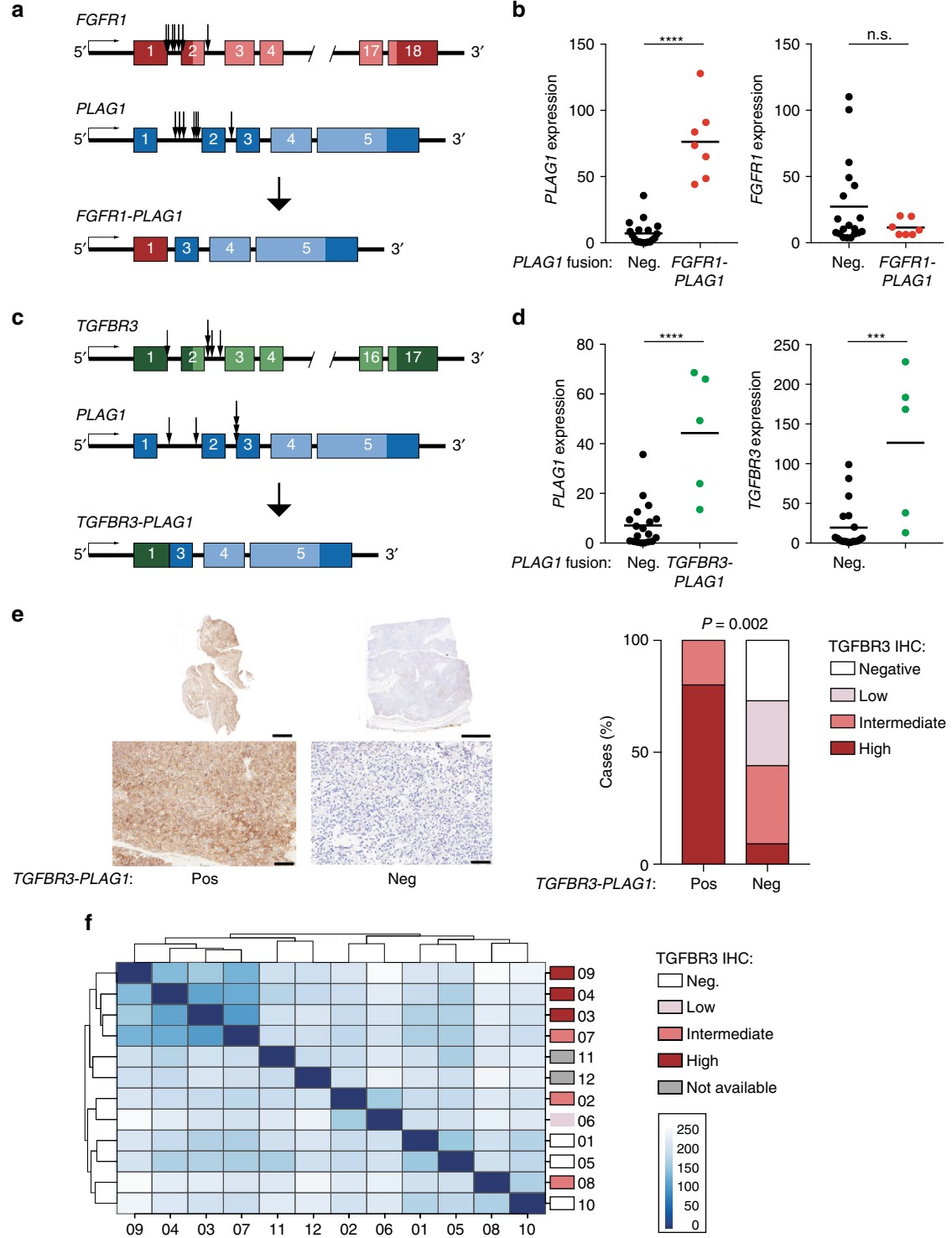

**Fig. 2** Detection of the *FGFR1-PLAG1* and *TGFBR3-PLAG1* fusion genes. **a** Illustration of the *FGFR1-PLAG1* fusion gene. Arrows show locations of genomic break points for each of the tumors. **b** Expression of *PLAG1* (left) and *FGFR1* (right) in *PLAG1* fusion-negative vs. *FGFR1-PLAG1*-positive tumors. Graphs show fragments per kilobase of exon per million fragments mapped (FPKM), based on RNA-seq data. ****$P < 0.0001$; Student's *t*-test. $n = 19 + 7$. Horizontal lines show mean values. **c** Illustration of the *TGFBR3-PLAG1* fusion gene. **d** Expression (FPKM) of *PLAG1* (left) and *TGFBR3* (right) in *PLAG1* fusion-negative vs. *TGFBR3-PLAG1*-positive tumors. ***$P < 0.001$, ****$P < 0.0001$; Student's *t*-test. $n = 19 + 5$. Horizontal lines show mean values. **e** Left, IHC showing TGFBR3 staining in whole tumor sections (upper panels, scale bars=5 mm) and enlargement of selected areas (lower panels, scale bars=100 μm). Right, quantification of IHC TGFBR3 levels in *TGFBR3-PLAG1* positive vs. negative tumors. $P$ (high TGFBR3) = 0.002, Fisher's exact test. $n = 5 + 34$. **f** Poisson sample clustering based on gene expression in cohort 1, annotated by TGFBR3 IHC results

140× for tumor DNA and 78× for normal DNA, and 97% of the target sequence was covered to at least 20× depth (Supplementary Fig. 1). Targeted resequencing showed a high validation rate (98%) of the detected mutations (see "Methods" section). RNA sequencing in the 12 cases resulted in an average of 132M reads per sample, of which 90% were aligned to the reference sequence with high quality (Supplementary Fig. 2). Twenty-eight additional formalin-fixed paraffin-embedded (FFPE) tumors (cohort 2) were analyzed with RNA sequencing with an average 138M reads per sample, of which 74% were aligned to the reference sequence with high quality (Supplementary Fig. 3). See Supplementary Figs. 4–6 for histology of the tumors, Supplementary Fig. 7 for diagnostic immunohistochemistry (IHC) results, and Supplementary Table 1 for clinical information.

We detected a median of 16 non-silent somatic mutations per tumor, corresponding to 0.5 mutations per megabase (Supplementary Fig. 8 and Supplementary Data 1). This mutational load is similar to that of salivary gland adenoid cystic carcinoma (0.3/MB)[12], and is lower than that of salivary duct carcinoma (1.7/MB) and most other adult solid cancers[13, 14]. Despite a low mutation count, MECA tumors showed evidence of intratumoral genetic heterogeneity, with the majority of sequenced tumors harboring at least 1 subclonal population (Supplementary Fig. 9).

One tumor had a hypermutated genotype, with 1116 somatic mutations and a transition/transversion ratio of 8.1. This was in contrast to the other cases, which had a mean transition/transversion ratio of 1.3. Three patients had hotspot mutations in genes involved in RAS signaling: $PIK3CA^{H1047L/R}$ in two cases, and a combination of $HRAS^{Q61R}$ and $PIK3CA^{K111E}$ in one case (Fig. 1, and Supplementary Data 1 and 2). These mutations appeared to be clonal events with high cancer cell fraction, consistent with early events in tumor development (Supplementary Fig. 9). Forty-eight percent of the somatic point mutations found in DNA were also detected by RNA sequencing (Supplementary Fig. 10A), indicating that they are expressed. The percentage of mutations detected by RNA sequencing was higher in cancer genes, and was independent of mutational load (Supplementary Fig. 10A–B). This is in line with previous findings in the head and neck squamous cell carcinoma and salivary duct carcinoma[14, 15].

**Fusion genes**. We detected fusion genes in 28 of 40 (70%) tumors (Fig. 1 and Supplementary Table 2); a majority of these were confirmed by RT-PCR (Supplementary Fig. 11) and/or FISH analysis (Supplementary Figs. 12 and 13). Fusion-positive cases had a lower mutation rate than fusion-negative ones (median: 12 vs. 40 mutations/tumor; $P = 0.0020$; Mann–Whitney test; Supplementary Fig. 8), and no mutations in cancer driver genes were found in fusion-positive tumors. The identified fusion genes were mutually exclusive with one another, consistent with a role as potential drivers of oncogenesis. These findings add MECA to the list of salivary malignancies that are known to harbor fusions, including adenoid cystic, mucoepidermoid, secretory, and hyalinizing clear cell carcinomas[16].

**PLAG1 translocations**. Half of MECA cases (21/40; 53%) harbored rearrangements involving the *PLAG1* oncogene, which was found in both MECA de novo and MECA ex-PA (Fig. 1 and Supplementary Table 2). PLAG1 is a transcription factor crucial for physiologic growth and development, and oncogenic *PLAG1* translocations leading to overexpression of the gene were previously mainly reported in PAs and lipoblastomas[17]. To compare the prevalence of *PLAG1* or 8q12 rearrangements between MECA and PA, we investigated an independent set of 442 PAs analyzed with karyotyping, and in selected cases array CGH and RT-PCR[18]

(and G.S., unpublished material). *PLAG1* rearrangements were present at a higher rate in MECAs compared to PAs (21/40, 53% vs. 102/442; 23%, OR=3.68; 95% confidence interval (CI) =1.91–7.12, $P = 10^{-4}$, Fisher exact test), indicating that enrichment of *PLAG1* rearrangements in MECA appears to be a hallmark of this salivary malignancy.

*FGFR1-PLAG1* **is enriched in MECA ex-PA**. *FGFR1-PLAG1* was detected in 7 of 40 (18%) tumors (Fig. 1). This fusion gene has been described previously in a small subset of PA[19], and co-existing rearrangements of *FGFR1* and *PLAG1* have also been detected in MECA ex-PA[6]. We found *FGFR1-PLAG1* in MECA ex-PAs only, supporting the hypothesis that the fusion gene was present in the PAs also before transformation into carcinoma in these cases. The *FGFR1-PLAG1* fusion was enriched 15-fold in MECA ex-PA compared to PA (7/24, 29.1%, 12/442, 2.7%; OR = 14.8, 95% CI=5.2–42.2, $P = 10^{-5}$, Fisher's exact test). Furthermore, the *FGFR1-PLAG1* fusion was not detected in 261 published and unpublished cases of other salivary gland cancers or in 20 cases of benign Warthin's tumor (Supplementary Table 3). The striking, specific enrichment of *FGFR1-PLAG1* in MECA ex-PA raises the possibility that this fusion in PA may predispose to malignant transformation, particularly along the MECA lineage.

The genomic break points were generally located between the promoter and the transcription start site of both *FGFR1* and *PLAG1*, potentially leading to expression of a full-length PLAG1 protein regulated by the *FGFR1* promoter (Fig. 2a). Indeed, the *FGFR1-PLAG1*-positive tumors showed high expression of *PLAG1* but low levels of *FGFR1* (Fig. 2b). IHC analysis showed positive nuclear PLAG1 staining in tumors positive for *FGFR1-PLAG1* (Supplementary Fig. 14).

In PA, the *FGFR1-PLAG1* fusions are generated by a ring chromosome, consisting of a small centromeric portion of chromosome 8[19]. We performed array CGH analysis and found a distinct amplification of chromosome 8 between the *FGFR1* and *PLAG1* loci, consistent with ring chromosome formation, r(8) (p12q12.1), in all seven MECA ex-PAs positive for *FGFR1-PLAG1* (Supplementary Fig. 15). To date, the *FGFR1-PLAG1* fusion has not been detected without ring formation.

**Detection of a novel and recurrent *TGFBR3-PLAG1* fusion**. In 6 of 40 (15%) tumors, we detected a *TGFBR3-PLAG1* fusion gene (Fig. 1 and Supplementary Table 2), which has not been previously reported in cancer. We found no evidence of *TGFBR3-PLAG1* or t(1;8) translocations in 442 PAs investigated by karyotyping, or in 281 published and unpublished cases of salivary gland tumors other than MECAs (Supplementary Table 4). Thus, available data indicate that the *TGFBR3-PLAG1* fusion appears to be specific for MECA.

Similar to what was noted for *FGFR1-PLAG1*, the genomic break points in the *TGFBR3-PLAG1* fusion were located before or just after the transcriptional start site of both *TGFBR3* and *PLAG1* (Fig. 2c). This could potentially lead to promoter swapping, and expression of full-length (or near full-length) PLAG1 and TGFBR3 proteins. As with *FGFR1-PLAG1*, *PLAG1* was significantly overexpressed in *TGFBR3-PLAG1*-positive tumors (Fig. 2d), and IHC analysis showed strongly positive nuclear PLAG1 staining (Supplementary Fig. 14). Notably, unlike other *PLAG1* rearrangements, tumors with *TGFBR3-PLAG1* showed overexpression also of the 5′ fusion partner, *TGFBR3* (Fig. 2d). IHC analysis confirmed high levels of TGFBR3 in *TGFBR3-PLAG1*-positive tumors (Fig. 2e).

Four of the six *TGFBR3-PLAG1*-positive cases were MECA de novo tumors (Fig. 1), and showed uniform and strong TGFBR3 staining. In one *TGFBR3-PLAG1*-positive MECA ex-PA,

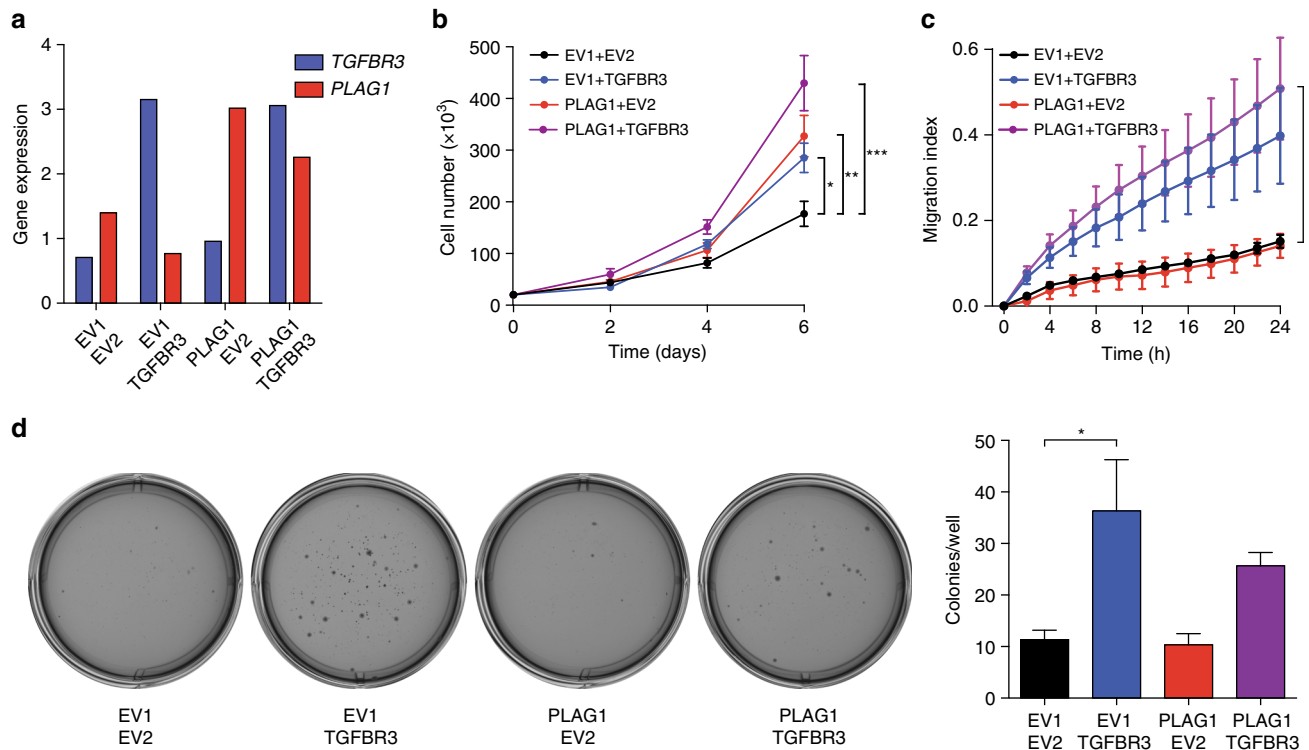

**Fig. 3** Overexpression of *TGFBR3* and *PLAG1* causes transformation of human salivary gland (HSG) cells. **a** RT-QPCR of *TGFBR3* and *PLAG1* mRNA levels in HSGs infected with constructs expressing combinations of *PLAG1* or the corresponding empty vector (*EV*) 1, and *TGFBR3* or the corresponding *EV2*. Data are mean using three technical replicates, normalized to the reference gene *STLM*. **b** Proliferation. Data are mean ± SEM using three technical replicates. *$P < 0.05$, **$P < 0.01$, ***$P < 0.001$; two-way ANNOVA with Dunnett's multiple comparison test. The experiment was replicated three times. **c** Migration index. Data are mean ± SEM using four technical replicates. *$P < 0.05$; two-way ANNOVA with Dunnett's multiple comparison test. The experiment was replicated two times. **d** Representative photographs (left) and quantification (right) of soft agar colony formation. Data are mean ± SEM from three technical replicates. *$P < 0.05$; one-way ANNOVA with Tukey's multiple comparison test. The experiment was replicated two times

IHC for TGFBR3 was uniformly positive in the carcinoma component, but negative in the PA component (Supplementary Fig. 16A). In the other MECA ex-PA case, *TGFBR3-PLAG1* was detected only in limited focal areas of the carcinoma component. The fusion-positive areas were the only parts of the tumor that were positive for TGFBR3 (Supplementary Fig. 16B). Together, these results suggest that the *TGFBR3-PLAG1* fusion gene causes overexpression of a functional TGFBR3 protein, likely contributing to the development of MECA, arising either in normal salivary gland tissue or a pre-existing PA.

We then asked whether high TGFBR3 expression is found in other types of salivary tumors. We examined 11 PAs, finding no cases with high TGFBR3 staining. In 23 non-MECA salivary cancers, however, we did identify high TGFBR3 staining in a subset of adenoid cystic carcinoma (ACC), epithelial myoepithelial carcinoma, and polymorphic low-grade adenocarcinoma samples (Supplementary Fig. 17). These data indicate that high TGFBR3 expression is found across multiple salivary cancer histologies, but not in benign adenomas, and that MECAs appear to achieve this overexpression via *TGFBR3-PLAG1* fusion.

We performed unsupervised Poisson clustering on MECA transcriptome data, and observed that the highly TGFBR3-positive cases clustered together in each cohort (Fig. 2f and Supplementary Fig. 18). This indicates that TGFBR3 overexpression is associated with a distinct transcriptional program in MECAs. In contrast, the *FGFR1-PLAG1* fusion did not cluster together in either cohort.

TGFBR3 can act as a tumor suppressor, which acts negatively on TGF-β signaling by binding to TGFBR1 and TGFBR2 separately, thereby inhibiting the TGFBR1/2 complex

formation[20]. On the other hand, overexpression of TGFBR3 has oncogenic potential in breast and colorectal cancer[21, 22]. We detected no significant difference in TGF-β signaling activity between tumors with high and low TGFBR3 levels, as judged by expression of 84 key genes responsive to TGF-β signal transduction (Supplementary Fig. 19). In line with this, the TGF-β signaling pathway activity was similar between the groups based on Ingenuity Pathway Analysis, which instead revealed altered activity of several cell cycle regulation pathways in TGFBR3 overexpressing tumors (Supplementary Fig. 20). These data suggest that the oncogenic effect of TGFBR3 overexpression in MECA is mediated by mechanisms independent of TGF-β signaling, which is in line with previous findings in breast cancer[20].

**Oncogenic potential of TGFBR3 and PLAG1 overexpression.** Overexpression of PLAG1 has been shown to have oncogenic potential, both in vitro and in vivo[23, 24]. However, the role of high TGFBR3 levels in salivary gland cancer has not been investigated. TGFBR3 has been described as having both oncogenic and tumor suppressive phenotypes, depending on context[21, 25]. To determine the effects of TGFBR3 and PLAG1 upregulation in salivary cells, we infected human salivary gland (HSG) cells with constructs leading to overexpression of PLAG1 and/or TGFBR3 at levels similar to those detected in *TGFBR3-PLAG1*-positive tumors (Fig. 3a). Overexpression of PLAG1 or TGFBR3 caused a similar increase in proliferation, whereas co-expression of TGFBR3 and PLAG1 led to a further increase suggesting an additive effect (Fig. 3b). Furthermore, overexpression of TGFBR3, but not of

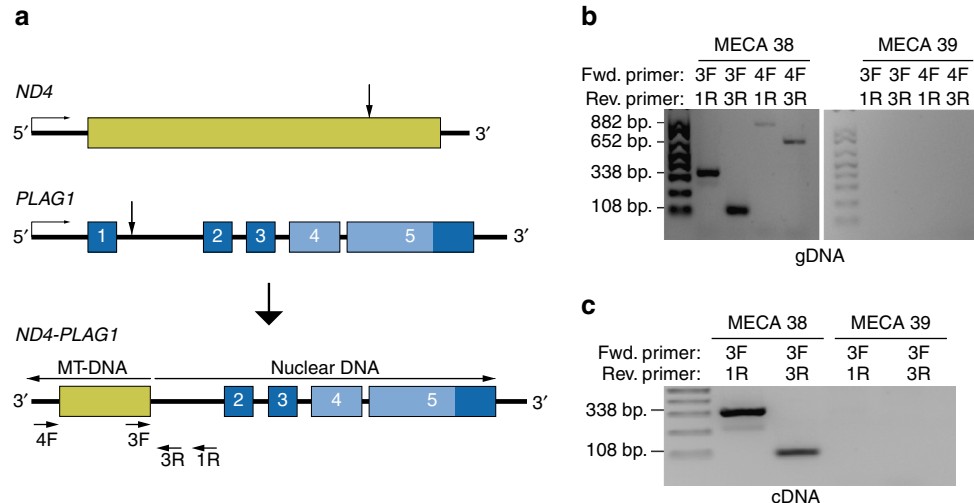

**Fig. 4** Detection of the novel mitochondrial/genomic DNA fusion gene *ND4-PLAG1*. **a** Illustration of *ND4-PLAG1*. MT mitochondrial. **b** PCRs of genomic DNA extracted from the *ND4-PLAG1*-positive tumor (case 38) and a fusion-negative tumor (case 39), using the denoted primer pairs. Of the 882 bp PCR product of primers 4F and 1R, 591 bp are mitochondrial and 291 bp are nuclear DNA. **c** PCRs of cDNA extracted from tumor 38 and 39

PLAG1, led to increased migration and soft agar colony formation in these cells (Fig. 3c, d).

We then examined the effect of TGFBR3 expression in non-malignant salivary cells. We overexpressed TGFBR3 in TCG580, an immortalized PA cell line that harbors a *PLAG1* rearrangement (a t(8;15) translocation with unknown fusion partner) but normal TGFBR3. Modest ectopic expression of TGFBR3 caused increased proliferation and migration (Supplementary Fig. 21A–C), but did not induce soft agar colony formation in these cells. We similarly examined cells from two primary PAs (C763, harboring a *CTNNB1-PLAG1* fusion, and C1329, negative for *PLAG1* rearrangements). Overexpression of *TGFBR3* and/or *PLAG1* caused continuous proliferation of primary cells, which otherwise senesced in vitro (Supplementary Fig. 22). Taken together, these data from malignant and non-malignant salivary cell lines suggest that the *TGFBR3-PLAG1* fusion, by promoting overexpression of each gene, is tumorigenic.

**ND4-PLAG1 resulting from a fusion between nuclear and mitochondrial DNA.** One case of MECA ex-PA harbored the novel fusion gene *ND4-PLAG1* (Fig. 4a). *ND4* is a mitochondrial gene that encodes the protein NADH dehydrogenase subunit 4. In this fusion, the 3′ part of *PLAG1*, including the full coding region of the gene, was fused with the inverted 3′ end of *ND4*. The rearrangement was detected by RNA sequencing (Supplementary Fig. 23), and was confirmed by PCR and Sanger sequencing of purified PCR products from genomic DNA and cDNA (Fig. 4b, c and Supplementary Fig. 24A). By using PCR primers with increasing distance from the break point, we determined that the size of the mitochondrial insertion was at least 591 bp, including not only *ND4* but also three downstream genes encoding transfer RNA (Supplementary Fig. 24B). Larger PCR products were not detected, which may reflect limited size of the mitochondrial DNA insertion, or may be due to degradation of the DNA as it was extracted from FFPE material. The *ND4-PLAG1*-positive case showed overexpression of *PLAG1* at similar levels as tumors with other *PLAG1* rearrangements, suggesting oncogenic potential of the fusion gene. Although insertions of mitochondrial DNA into the nuclear genome have been detected in a small subset of human cancers[26], this is the first report of an oncogenic fusion gene formed by a junction of mitochondrial and nuclear DNA. The mechanism by which

mitochondrial DNA entered the nucleus and fused with chromosome 8 in this case remains unknown.

**Other PLAG1 fusions.** In total, *PLAG1* rearrangements were found in 21 (53%) of the tumors (Fig. 1). Other than *FGFR1* and *TGFBR3*, each *PLAG1* fusion partner was identified in one case. One MECA ex-PA harbored *CTNNB1-PLAG1* (Supplementary Fig. 25A), which was previously reported in PA[8, 10, 11, 27]. We also detected the novel fusion genes *ACTA2-PLAG1*, *GEM-PLAG1*, and *NCALD-PLAG1* (Supplementary Fig. 25B–D). In all these cases, the break points were located between the promoter and the transcription start site of both *PLAG1* and the 5′ fusion partner, potentially leading to ectopic expression of full-length PLAG1 under control of the *GEM*, *ACTA2*, or *NCALD* promoters, respectively.

One tumor harbored *NKTR-PLAG1*, where *PLAG1* was fused with intron 10 of the inverted *NKTR* gene (Supplementary Fig. 25E). In another case, *PLAG1* was fused with an intergenic region located only 600 kb away from *PLAG1*, likely resulting from a deletion in the chromosome region 8q12. Similar to the *ND4-PLAG1* fusion gene, these rearrangements are not likely to cause promoter swapping. Instead, they might mediate overexpression of *PLAG1* by locating regulatory elements such as enhancer regions upstream of the gene. In two tumors, RNA sequencing showed evidence of a *PLAG1* fusion gene, but we were not able to determine the fusion partner.

**Effector pathway activation in PLAG1 fusion-positive tumors.** All tumors with a *PLAG1* rearrangement showed overexpression of *PLAG1*, and *FGFR1-PLAG1*-positive cases tended to have the highest *PLAG1* levels (Supplementary Fig. 26A). The other PLAG1 family members, *PLAGL1* and *PLAGL2*, were equally expressed in *PLAG1* fusion-positive and -negative cases (Supplementary Fig. 26B). The oncogenic effect of PLAG1 overexpression has been hypothesized to be mediated by different mechanisms, such as activation of insulin-like growth factor 2 (IGF2) and possibly also Wnt signaling[23, 28]. We found that *PLAG1* rearrangements were associated with increased expression of IGF2, and of the other putative *PLAG1* effector genes CRABBP2 and CRLF1 (Supplementary Fig. 26C). On the other hand, Wnt signaling, estimated by analyzing the expression of five known effector genes[29], did not appear to differ between *PLAG1*

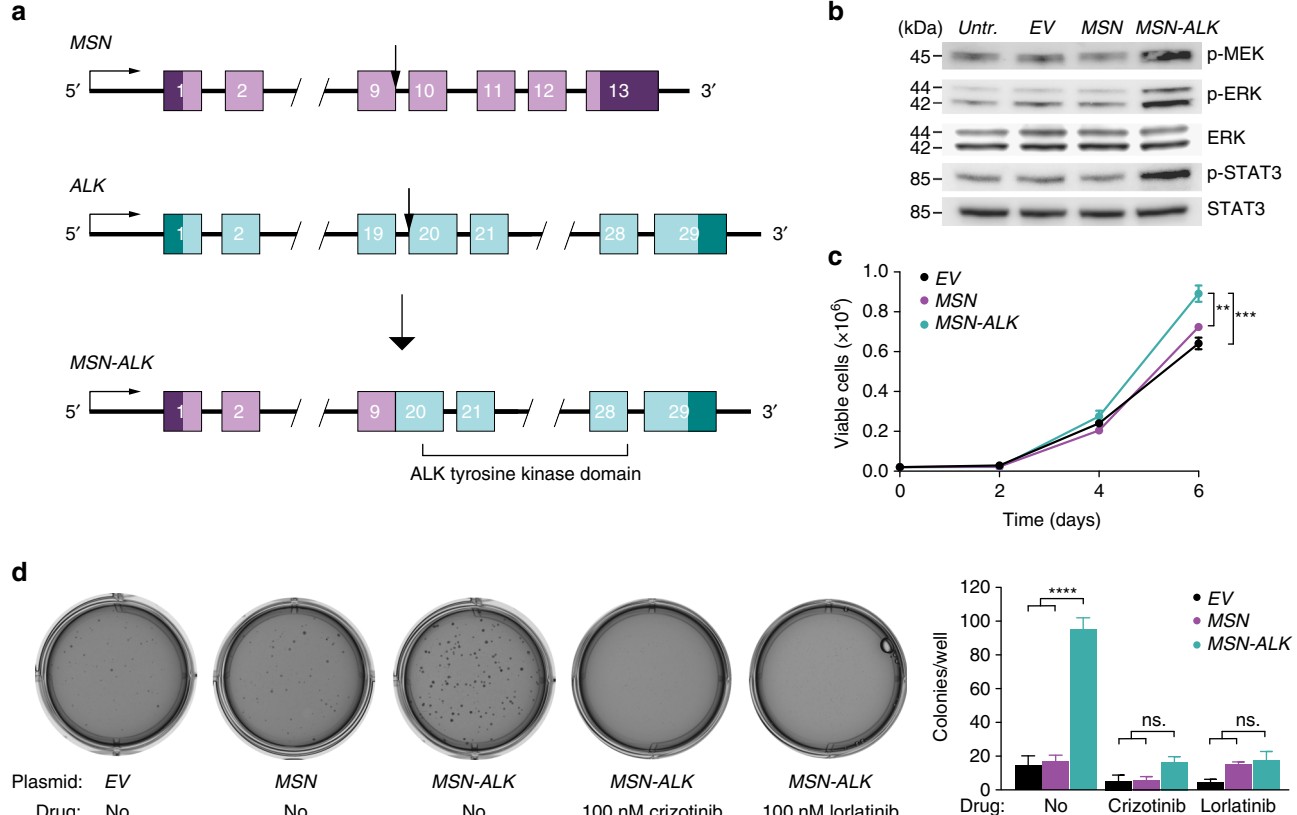

**Fig. 5** *MSN-ALK* is an activating and oncogenic fusion gene potentially targetable with ALK inhibitors. **a** Illustration of the *MSN-ALK* fusion gene. Arrows show genomic break points. **b** Western blot showing levels of ALK downstream signaling proteins in human salivary gland (HSG) cells expressing empty vector (*EV*), *MSN*, or *MSN-ALK*. P phosphorylated, kDa kilodalton. **c** Proliferation of HSG cells expressing *EV*, *MSN*, or *MSN-ALK*. Data are mean ± SEM using three technical replicates. **P < 0.01, ***P < 0.001; two-way ANOVA with Tukey's multiple comparison test. The experiment was replicated two times. **d** Representative photographs (left) and quantification (right) of soft agar colony formation assay. Data are mean ± SEM using three technical replicates. ****P < 0.0001; one-way ANOVA with Tukey's multiple comparison test. The experiment was replicated two times

fusion-positive and -negative cases (Supplementary Fig. 26D). This could potentially suggest that IGF2, but not Wnt, may be involved in the oncogenic effects of *PLAG1* fusion genes in MECA.

**HMGA2 rearrangements**. *HMGA2* encodes a small chromatin-binding protein that mediates transcriptional activation of several oncogenic functions, including inhibition of DNA repair and p53-induced apoptosis[30]. Rearrangements affecting *HMGA2* occur in PA as well as carcinoma ex-PA[6, 7, 9, 10, 31], and are also found in other benign tumor types such as uterine leiomyomas and lipomas[17, 32]. Although *HMGA2-WIF1* has been reported in PA[27, 31, 33], other *HMGA2* fusion partners in salivary gland tumors remain largely unknown[6, 11, 34].

We detected *HMGA2* rearrangements in one MECA de novo and three MECA ex-PA tumors, which all showed overexpression of *HMGA2*. The genomic break points were located within the 3′ untranslated region of *HMGA2*, which was fused with intergenic regions from chromosomes 3, 6, or 12 (Supplementary Fig. 27A–C). The finding that *HMGA2* fuses with apparently random genomic locations suggests that the fusion partner of *HMGA2* may be irrelevant in MECA. This is in line with the hypothesis that loss of target sites for the negatively regulating miRNA let-7 leads to overexpression of full-length HMGA2[35, 36]. Of note, two of the four tumors with HMGA2 rearrangements also harbored an *FGFR2*[M186T] mutation, raising the possibility that these events may cooperate in tumor development.

**EWSR1-ATF1 in MECA de novo tumors**. Fusions of the 5′ part of EWS RNA-binding protein 1 (*EWSR1*) and the 3′ part of activating transcription factor 1 (*ATF1*) are found in a majority of salivary gland hyalinizing clear cell carcinomas[37], and were also reported in soft tissue myoepithelial tumors[38]. We detected *EWSR1-ATF1* in two de novo tumors (Supplementary Fig. 28A), and re-review by a salivary gland pathologist (N.K.) revealed typical MECA morphology as well as clear cell features in both of these cases (Supplementary Fig. 28B). They were therefore considered clear cell MECA, a variant that has been described previously[39].

**Characterization of MSN-ALK**. Anaplastic lymphoma receptor tyrosine kinase (ALK) translocations are found in several tumor types, and are an important target for treatment with ALK inhibitors in patients with non-small cell lung cancer[40]. One MECA de novo tumor harbored a fusion between exon 9 of moesin (*MSN*) and exon 20 of *ALK* (Fig. 5a). *MSN-ALK* fusions have only previously been detected in anaplastic large cell lymphoma[41]. Whereas 16 other *ALK* fusion genes have been shown to activate the ALK tyrosine kinase domain[42], the effect of *MSN-ALK* on downstream signaling remains unknown.

To test the biological impact of *MSN-ALK*, we infected HSG cells with plasmids expressing *MSN* or *MSN-ALK* cloned from the fusion-positive patient's tumor DNA, or an empty vector (*EV*). Expression of *MSN-ALK* was confirmed on DNA, RNA, and protein levels (Supplementary Fig. 29A–C), and led to increased phosphorylation of the downstream targets ERK and

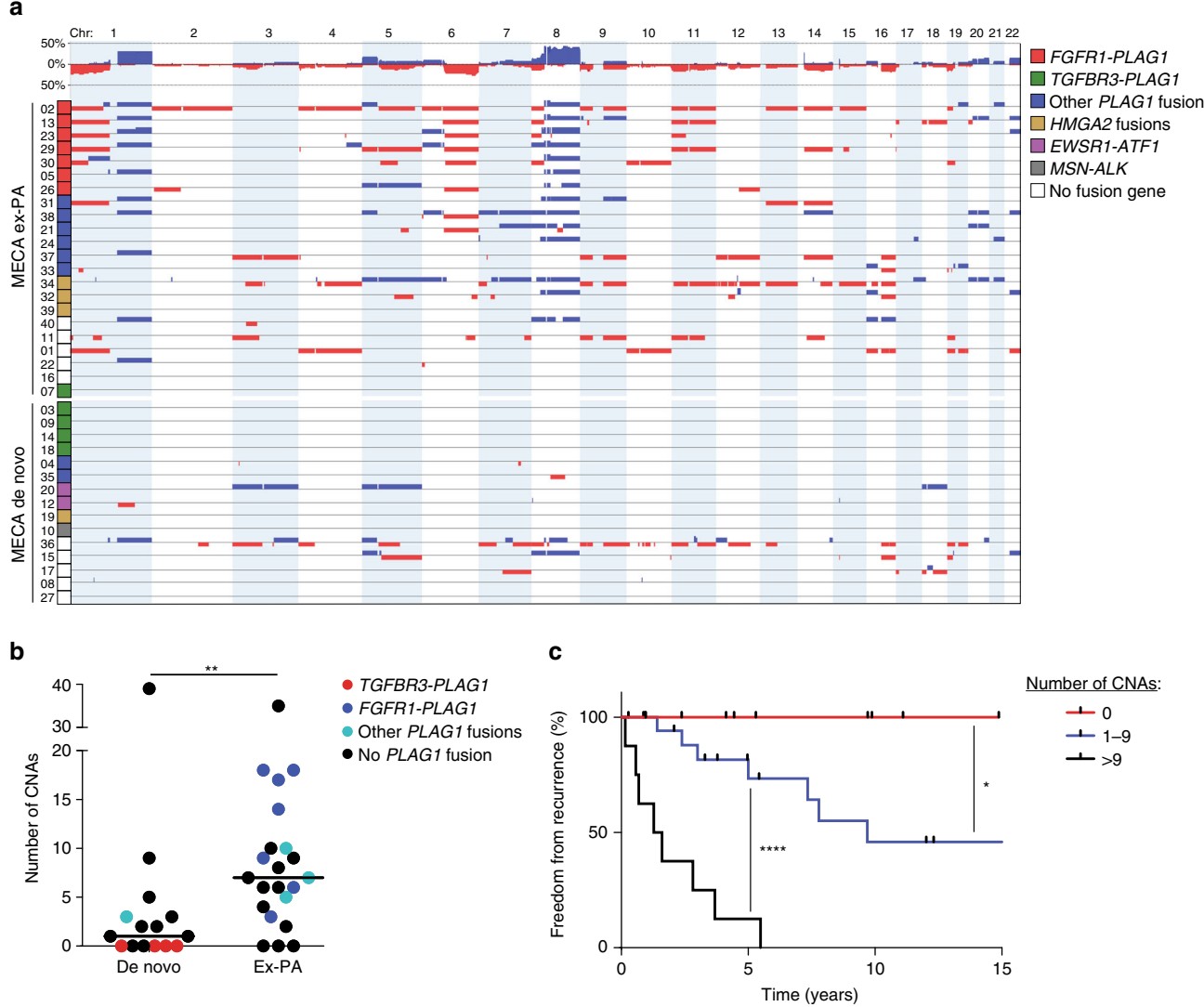

**Fig. 6** The number of copy number alterations is associated with tumor origin and prognosis. **a** All significant CNAs in MECA tumors, grouped by origin and fusion gene status. **b** Number of CNAs in MECA de novo and MECA ex-PA tumors. **P < 0.01; Mann–Whitney test. n = 15 + 22. **c** Freedom from recurrence in patients with different number of CNAs. *P < 0.05, ****P < 0.0001; log-rank test. n = 9 (0 CNAs) + 20 (1–9 CNAs) + 8 (>9 CNAs)

STAT3 (Fig. 5b). These findings were confirmed using HEK cells (Supplementary Fig. 30A–C). Unlike overexpression of *MSN*, *MSN-ALK* also stimulated proliferation and soft agar colony formation in HSG cells (Fig. 5c–d). Treatment with the ALK inhibitors crizotinib (100 nM) or lorlatinib (100 nM) reduced the colony formation of cells expressing *MSN-ALK* to baseline levels (Fig. 5d). Together, these results show that *MSN-ALK* is an activating and oncogenic fusion gene, and that *MSN-ALK*-positive tumors may potentially respond to ALK inhibitor treatment.

**Gene alterations leading to potential neoantigen formation.** Neoantigens are novel protein sequences resulting from genetic alterations in cancer cells which, if able to bind to MHC molecules on the surface of antigen-presenting cells, may then activate cytotoxic T lymphocytes. Tumors with a high mutational burden tend to be more likely to respond to checkpoint blockade immunotherapy[43, 44], an effect probably mediated by a higher number of neoantigens in these tumors[45]. MECAs have low mutational load. As expected, we identified a low number of mutation-associated neoantigens predicted to bind to MHC with high affinity (median 2 per tumor; Supplementary Fig. 31). We

then investigated the potential of fusion genes to generate neoepitopes. Whereas no potential neoantigens were found in the *FGFR1-PLAG1*-positive tumors, the *TGFBR3-PLAG1* fusions resulted in several neoepitopes with predicted high-affinity binding to MHC molecules (Supplementary Table 5). These findings are based on in silico predictions only and should be interpreted with caution, but indicate that, even in the presence of low mutation count, recurrent gene fusions generate neoepitopes that may have relevance for neoantigen-targeted immunotherapy.

**Copy number alterations.** Array CGH analysis revealed a higher number of copy number alterations (CNAs) in MECA ex-PAs compared to MECA de novo tumors (P = 0.0036; Fig. 6a, b and Supplementary Data 3). The most prevalent CNAs were amplifications of regions on chromosome 8 and gain of chromosome 1q, both of which were predominantly found in MECA ex-PA tumors (Supplementary Table 6).

Some of the tumors showed a recurrent pattern of CNAs. For example, four tumors positive for *FGFR1-PLAG1* all had chromosome 1p deletion, 1q gain, 6q deletion, 8q amplification, and 11p deletion, which may suggest combinatorial oncogenic effects of these alterations (Fig. 6a).

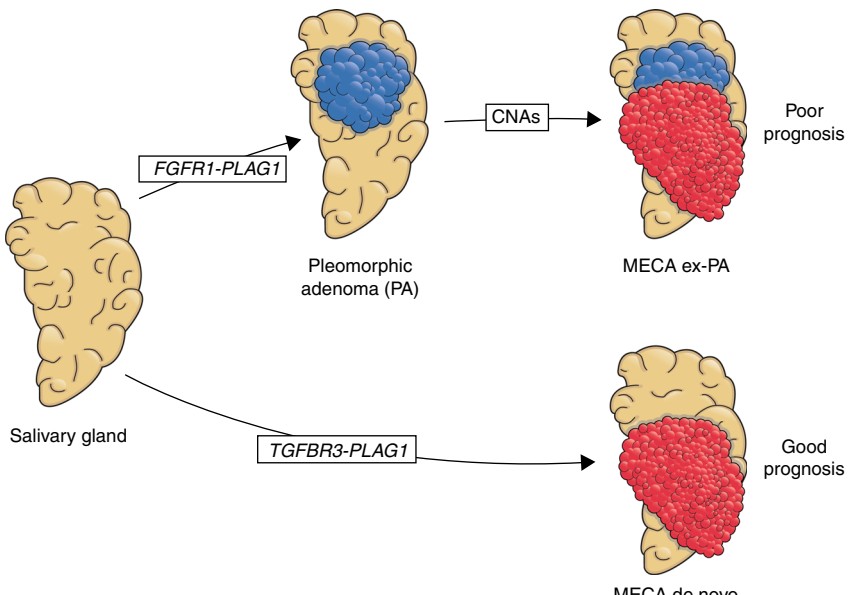

**Fig. 7** The role of *FGFR1-PLAG1* and *TGFBR3-PLAG1* in MECA development

Chromosome 6q deletions have been reported in several types of salivary cancer[46], and were found in eight (20%) of the MECAs. The detection of 6q deletions was associated with a higher total number of CNAs (mean 11.5 vs. 6.1 CNAs/tumor, $P = 0.0029$, Mann–Whitney test), which is in line with previous findings in ACC[47]. The PLAG-like 1 (*PLAGL1*) transcription factor has been suggested as a putative tumor suppressor gene that may mediate the oncogenic effect of chromosome 6q deletions[48], but previous studies did not show reduced *PLAGL1* expression in ACCs with 6q translocations[49, 50]. We detected low *PLAGL1* expression in four of eight tumors with 6q deletion (Supplementary Fig. 32A), but the significance of *PLAGL1* in MECA remains unknown.

Amplification of 8q has been previously reported in MECA but is not prevalent in benign myoepitheliomas[51]. We detected 8q amplifications/gains in all *FGFR1-PLAG1*-positive tumors and in a majority of MECA ex-PA tumors with other *PLAG1* fusion genes (Fig. 6a). As r(8) amplification in *FGFR1-PLAG1*-positive tumors only include a small portion of the q arm, we hypothesize that gains of 8q may mediate additional oncogenic effects in tumors with r(8) formation.

Homozygous or heterozygous loss of the 9p21–22 region, including the *CDKN2A* tumor suppressor gene, were reported in soft tissue myoepithelial tumors[52]. We found localized homozygous loss of the *CDKN2A* region in two cases, with only 13 and 37 genes included in the deletions, respectively (Supplementary Table 6). These tumors also showed low expression of *CDKN2A*, unlike four other cases that harbored heterozygous loss of the entire chromosome 9 (Supplementary Fig. 32B). Thus, as in other myoepithelial tumors, loss of *CDKN2A* may contribute to malignant transformation in a small subset of MECAs.

To further investigate the role of CNAs in MECA development, we also analyzed 17 PAs using array CGH. In stark contrast to MECA, PAs had a very quiet copy number profile. We detected only one CNA in all PAs combined (Supplementary Table 7), which was significantly fewer than in MECA ex-PA tumors (mean 0.06 vs. 8.8 CNAs/tumor, $P < 0.0001$, Mann–Whitney test). We hypothesize that CNAs are an important mediator of the malignant transformation of PAs into MECA ex-PA.

Interestingly, no CNAs were found in *TGFBR3-PLAG1*-positive tumors. Since this fusion gene was detected only in

MECA de novo cases or in the malignant component of MECA ex-PA tumors, this mutual exclusivity suggests that *TGFBR3-PLAG1* is a potent oncogenic event that leads to carcinoma development even in the absence of CNAs. Conversely, other *PLAG1* fusion genes were more commonly found in MECA ex-PA tumors and were associated with a high number of CNAs. We hypothesize that these *PLAG1* fusions cause PA, and that the transformation into MECA ex-PA may be mediated by CNAs in these cases.

**Clinical correlates**. Among patient clinical characteristics, male sex was associated with a higher risk of tumor recurrence (Supplementary Fig. 33). Other factors, including tumor origin (MECA de novo vs. ex-PA), primary tumor site (major vs. minor salivary glands), smoking history, or tumor stage at diagnosis, did not have prognostic relevance in this cohort.

We investigated the prognostic role of different genetic alterations in MECA. Most notably, higher number of CNAs was associated with poorer prognosis. Patients with 10 or more CNAs had a median recurrence-free time of 17 months, and all these tumors resulted in recurrence within 6 years. On the other hand, none of the tumors without significant CNAs had recurred after a median follow-up time of 5 (range, 1–15) years (hazard ratio >9 CNAs vs. no CNAs = 20.6, 95% CI=4.6–91.0, $P < 0.0001$, log-rank test). In line with this finding, patients with 1–10 CNAs had an intermediate prognosis (hazard ratio 1–9 CNAs vs. no CNAs = 4.6, 95% CI=1.1–19.8, $P = 0.04$, log-rank test), and experienced recurrence in around 50% of the cases (Fig. 6c). A correlation between recurrence risk and number of CNAs was seen also within different tumor stage categories at diagnosis (Supplementary Fig. 34). Together, these findings indicate that genomic instability confers a poor prognosis in MECA. On the other hand, detection of any fusion gene, or any *PLAG1* fusion gene, did not correlate with survival or tumor stage at diagnosis in these patients.

**Discussion**

We performed a comprehensive analysis of the genome and transcriptome of MECA, an understudied, lethal salivary cancer. We found that MECA exhibits a molecular profile similar to

certain soft tissue tumors and sarcomas[53], in that they are defined by a low mutational load and frequent oncogenic gene fusions, and/or widespread arm-level CNAs acting as the main driving events (e.g., similar to pleomorphic sarcomas). Despite these similarities with soft tissue tumors, MECA appears molecularly distinct from myoepithelial tumors of soft tissue, which are morphologically similar tumors. Most notably, whereas *EWSR1* translocations are frequently detected in soft tissue myoepithelial tumors[54], *PLAG1* translocations are more common in MECAs.

Cancer driver gene mutations (e.g., *PIK3CA*, *HRAS*) were identified in only a minority of tumors, and the average mutational load was low at 0.5/MB. The majority (70%) of MECAs harbored fusion genes, many of which appear to be tumorigenic, either through upregulation of PLAG1 and TGFBR3 together (via *TGFBR3-PLAG1*), *MSN-ALK* or *EWSR1-ATF1*. These translocations were only identified in MECA de novo tumors or were restricted to the malignant component of MECA ex-PAs, suggesting that they mediate a potent oncogenic transformation of normal salivary gland or benign PA cells.

On the other hand, *FGFR1-PLAG1* fusions were only found in MECA ex-PAs. This fusion exists in a small proportion of benign PAs, which may then undergo malignant transformation in the context of the fusion combined with genomic instability manifested as complex copy number alteration (Fig. 7). These tumors are associated with a poor prognosis, which may be attributed to the higher degree of genomic instability.

The classification of MECA de novo or ex-PA tumors was based on histological findings only. Therefore, we cannot exclude the possibility that some of the tumors designated as MECA de novo may have in fact originated from PAs that were subsequently overgrown and obscured by the carcinoma component and therefore not detected histologically, as has been suggested previously in salivary duct carcinoma[55].

The findings of this study may have an impact on the clinical management of MECA. Diagnosing MECA remains challenging because of the morphological similarities with other types of salivary gland cancer. As *TGFBR3-PLAG1* has not been reported in other tumor types, this fusion may be a potential molecular marker of MECA. In addition, several of the genetic events are potentially targetable, e.g., by inhibiting downstream effectors of PLAG1 or targeting ALK. While low tumor mutational load is anticipated to be associated with lower degrees of response to immune checkpoint inhibitors[43, 44], recurrent fusion genes that result in novel transcripts may result in immunogenic neoepitopes that would represent attractive targets for neoantigen-based vaccines. Finally, the correlation between CNAs and poorer prognosis may be of value for future clinical decisions regarding treatment and monitoring of MECA patients.

## Methods
**Patients**. After obtaining written informed consent and approval from the Institutional Review Board, 40 patients diagnosed with myoepithelial carcinoma (MECA) of the salivary gland were treated at the Memorial Sloan Kettering Cancer Center (MSKCC) during the years 1965–2014 were included in the study. Clinical data from 34 patients were previously reported[5]. Eight patients were included in previous studies examining *PLAG1* and *HMGA2* rearrangement status[6, 34].

**Diagnostic procedure**. Upon inclusion in the study, hematoxylin and eosin (H/E) stained sections were independently re-evaluated by two head and neck pathologists (N.K. and R.G.). The inclusion criteria for IHC were positive staining of at least one keratin marker in combination with positive staining of at least one myoepithelial marker or S100. In cases that initially did not meet these criteria due to lacking IHC results, we performed additional IHC to confirm the diagnosis.

**DNA and RNA extraction**. For cohort 1 (cases 01–12), tissue samples were snap frozen in liquid nitrogen at the time of surgery and stored at −80 °C. Tumor and matched normal (from peripheral blood or non-neoplastic tissue) DNA and tumor RNA were extracted using the DNeasy Blood and Tissue Kit (Qiagen, Venlo, the

Netherlands) and the RNeasy Plus Mini Kit (Qiagen), respectively. For cohort 2 (cases 13–40), FFPE tumor tissue was sectioned and stored at room temperature. DNA and RNA were extracted using the QIAamp DNA FFPE Tissue Kit (Qiagen) and the RNeasy FFPE kit (Qiagen), respectively. DNA was quantified with Nanodrop 8000 (Thermo Fisher Scientific, Waltham, MA, USA) and the PicoGreen assay (Thermo Fisher Scientific). RNA was quantified with the RiboGreen assay (Thermo Fisher Scientific).

**Whole-exome sequencing**. Libraries were generated from ~1.5 μg genomic DNA using the SureSelect XT Library Preparation Kit (Agilent Technologies, Santa Clara, CA, USA). DNA was fragmented with an LE220 Focus Ultra-sonicator (Covaris, Woburn, MA, USA), end repaired using the End-It kit (Epicentre, Madison, WI, USA), adenylated, ligated to sequencing adapters (Illumina, San Diego, CA, USA), and amplified by PCR. Exome libraries were captured with the SureSelect XT v4 51 Mb capture probe set (Agilent Technologies), enriched by PCR, and quantified using the KAPA Library Quantification Kit (KAPA Biosystems, Wilmington, MA, USA), 2100 BioAnalyzer (Agilent Technologies), and Qubit Fluorometer (Thermo Fisher Scientific). Sequencing was done on a HiSeq2500 (Illumina) using 2 × 125 bp cycles.

**RNA sequencing**. In cohort 1, RNA sequencing libraries were generated from ~100 ng total RNA extracted from fresh frozen tumor tissue, using the KAPA Stranded RNA-Seq with RiboErase sample preparation kit (KAPA Biosystems). RNA was subjected to ribosomal depletion, first- and second-strand cDNA synthesis, adapter ligation, and PCR amplification using 11 cycles. Libraries were quantified using the KAPA Library Quantification Kit (KAPA Biosystems), 2100 BioAnalyzer (Agilent Technologies), and Qubit Fluorometer (Thermo Fisher Scientific), and were sequenced on a HiSeq2500 (Illumina) using 2 × 125 bp cycles. In cohort 2, libraries were generated from ~1 μg RNA extracted from FFPE, using the TruSeq® Stranded Total RNA LT Kit (Illumina). PCR was amplified using six cycles, and sequencing was done using 2 × 50 bp cycles on a HiSeq2500.

**Mutation analysis**. Raw whole-exome seq data were aligned to the hg19 reference sequence using the Burrows-Wheeler Aligner (BWA) version 0.7.10[56]. Realignment of insertions and deletions (indels), base quality score recalibration, and duplicate reads removal were done with the Genome Analysis Toolkit (GATK) version 3.2.2 (broadinstitute.org/gatk), according to the raw read alignment guidelines[57]. Single nucleotide variants (SNVs) were independently detected by MuTect[58], Somatic Sniper v1.0.4.2[59], Strelka v1[60], and Varscan v2.3.8[61], and significant SNVs were identified using an in-house pipeline[14]. In brief, SNVs that were identified by ≥2 callers, with ≥8× coverage and ≥10% variant allelic fraction in tumor DNA, as well as ≥15× coverage and >97% normal allelic fraction in normal DNA, were considered high-confidence variants. Variants that met the criteria for tumor coverage and variant allelic fraction but were detected by one caller only and/or had a normal DNA coverage of 4–14×, and passed manual review using the Integrative Genomics Viewer (IGV) v2.3 (broadinstitute.org/igv) were considered low-confidence SNVs. Indels were detected by Strelka and VarScan. Variants with ≥4× tumor allelic coverage, >10% tumor allelic fraction, ≥4× normal DNA coverage, and >97% normal allelic fraction that passed manual review in IGV were considered potential indels.

In cohort 2, mutations were called from RNA-seq data using HaplotypeCaller. Since no normal tissue DNA or exome seq data was available in this cohort, only mutations affecting genes listed in COSMIC Cancer Gene Census (http://cancer.sanger.ac.uk/cencus), and present in the COSMIC database (http://cancer.sanger.ac.uk/cosmic), were evaluated. Variants with ≥15× coverage and ≥20% tumor allelic fraction that passed manual review using IGV were considered as true mutations.

**Validation of mutations**. The match between tumor and normal samples of each patient was confirmed with VerifyBamID[62], and with fingerprinting analysis using an in-house panel of 118 single nucleotide polymorphisms. In cohort 1, a random selection (15%) of the high-confidence SNVs, as well as all potential indels and low-confidence SNVs, was subjected to orthogonal validation using NimbleGen SeqCap EZ target enrichment (Roche, Basel, Switzerland), with a 500× and 250× sequencing depth for tumor and normal DNA, respectively. Variants achieving >100× coverage of both tumor and normal DNA, >15% variant allelic fraction in tumor, and <3% variant allelic fraction in normal DNA, were considered validated. The overall validation rate in high-confidence SNVs was 98%. All high-confidence SNVs, as well as validated indels (80% of all potential indels) and low-confidence SNVs (20% of all), were considered true mutations.

**Detection of fusion genes**. To detect fusion genes, RNA-seq bam files were analyzed using the software tools FusionCatcher (http://biorxiv.org/content/early/2014/11/19/011650) and deFuse[63]. Potential fusion genes were manually confirmed by reviewing the break point loci of each gene using IGV v2.3.

**Array comparative genomic hybridization (array CGH)**. Array CGH analysis was performed using the human genome CGH 4 × 180 K microarray (G4449A;

Agilent Technologies). Approximately 500 ng of tumor and reference DNA were fragmented, labeled, hybridized, and washed according to the manufacturer's instructions (Agilent Technologies). The arrays were subsequently scanned on a high-resolution C Microarray Scanner and the images processed using Feature Extraction v.10.7.1 (Agilent Technologies) with linear normalization (protocol CGH_107_Sep09). Data were analyzed using Nexus Copy Number Software® Discovery Edition v. 8.0 (BioDiscovery, El Segundo, CA, USA). CNAs were defined using the FASST2 segmentation algorithm with a significance threshold of $P = 1.0E$ −8. The $\log_2$ ratio thresholds for aberrations calls were set to 1.5 for amplification, 0.3 for gain, −0.3 for loss, and −1.5 for homozygous deletion. Each aberration was manually validated to confirm the accuracy of the call and excluded when occurring in sex chromosomes or in regions reported to have a high prevalence of copy number variation (database of genomic variants; http://dgvbeta.tcag.ca/dgv/app/news?ref=NCBI37/hg19).

**Gene expression analysis**. RNA-seq FASTQ files were aligned to the hg19 genome using STAR aligner with default parameters[64], counted using Rsamtools v3.2, (bioconductor.org/packages/release/bioc/html/Rsamtools.html) and annotated using the TxDb.Hsapiens.UCSC.hg19.known Gene version 3.2 transcript database. Regularized logarithm transformation of the matrix was obtained with the rlog function of DeSeq2 v1.10.1 after adjusting for the batch effect using the sva package v3.18.0 (svaseq function). Fragments per kilobase of exon per million fragments mapped were obtained with DESeq2 using the robust method. Differences in signaling pathway activity between groups of tumors were analyzed using ingenuity pathway analysis (www.qiagenbioinformatics.com/products/ingenuity-pathway-analysis).

**Tumor clonality analysis**. The degree of intratumoral genetic heterogeneity in cohort 1 samples with at least 10 mutations ($n = 9$) was analyzed by integrating mutation allelic fractions, tumor purity, and allele-specific copy number. In brief, mutations were clustered by analyzing reference allele and variant allele read counts using PyClone[65]. The estimated tumor purity and the copy number of the major and minor alleles in each region were determined by FACETS[66]. PyClone output data were clustered using a Drichlet process, estimating the number of mutation-defined clusters/tumor[67].

**Prediction of neoantigen formation**. Neoepitopes resulting from gene fusions were integrated with predicted HLA alleles from HLAminer[68] by INTEGRATE-Neo[69]. For both fusion genes and mutations, NetMHC 4.0[70] was used for neoantigen-binding prediction.

**Immunohistochemistry**. Sections (5 μm) of FFPE tumor tissue were stained with antibodies specific for TGFBR3 (HPA008257, polyclonal rabbit, dilution 1:130; Sigma-Aldrich, St. Louis, MO, USA), PLAG1 (H00005324-M02, monoclonal mouse, 1:50; Novus Biologicals, Litleton, CO, USA), phospho-Smad2/3 (SC11769, polyclonal goat, 1:200; Santa Cruz Biotechnology, Dallas, TX, USA), S100 (Z0311, polyclonal rabbit, 1:8000; Dako, Glostrup, Denmark), epithelial membrane antigen (EMA) (790-4463, monoclonal mouse, ready to use; Ventana Medical Systems, Tucson, AZ, USA), cytokeratin clone CAM5.2 (349205, monoclonal mouse, 1:50; Becton Dickinson, Franklin Lakes, NJ, USA), cytokeratin clone AE1/AE3 (M3515, monoclonal mouse, 1:1600; Dako), cytokeratin 7 (M7018, monoclonal mouse, 1:800; Dako), anti-keratin (34βE12) (790–4373, monoclonal mouse, ready to use; Ventana), calponin (760–4376, monoclonal rabbit, ready to use; Ventana), smooth muscle actin (202 M, monoclonal mouse, 1:100; Biocare Medical, Pacheco, CA, USA), p63 (790–4509, monoclonal mouse, ready to use; Ventana), and glial fibrillary acidic protein (760–4345, monoclonal rabbit, ready to use; Ventana).

A head and neck pathologist specialized in salivary gland cancer (N.K.) blinded to the genotype of the tumors estimated TGFBR3 staining intensity, scored as intensity 1–3, and percent of tumor area with positive staining. Tumors were then grouped as high (≥80% positive area, intensity 2–3), intermediate (20–79% positive, intensity 2–3), low (1–19% positive, intensity 1–3 or ≥20% positive, intensity 1) or negative (0% positive) for TGFBR3.

**PCR analyses**. cDNA was synthesized from tumor RNA using Superscript III Reverse Transcriptase (Thermo Fisher Scientific). For validation of fusion genes detected by RNA-seq, cDNA was amplified by reverse transcriptase PCR (RT-PCR) using a Master Cycler Pro (Eppendorf, Hamburg, Germany) and visualized on a 1.5% agarose gel. See Supplementary Fig. 35 for full pictures of gels that were cropped in main figures. For gene expression analysis, cDNA was analyzed by quantitative PCR using a QuantStudio 6 Flex (Applied Biosystems, Waltham, MA, USA). The ΔΔCt values were calculated with the $2^{-(\Delta\Delta Ct)}$ method. The analyzed genes were normalized to the reference genes STLM (for cell culture experiments) or GAPDH (for analysis of tumor tissue). All primer sequences are listed in Supplementary Table 8.

**Fluorescence in situ hybridization (FISH)**. FISH was performed on paraffin section using dual-color, dual-fusion translocation probes (see Supplementary Table 9 for clone list). Probe labeling, tissue processing, hybridization, post-

hybridization washing, and fluorescence detection were performed according to standard procedures. Slides were scanned using an Axioplan 2i epifluorescence microscope (Zeiss, Oberkochen, Germany) equipped with a 6 megapixel CCD camera (CV-M4 + CL, JAI) controlled by Isis 5.5.9 imaging software (MetaSystems Group Inc, Waltham, MA, USA). Marked region(s) within the section were scanned through 63× or 100× and at least five images per representative region captured (each image was a compressed stack of 12 or 15 z-section images taken at 0.5 micron intervals). Signal counts were performed on the captured images and a minimum of 30 discrete nuclei scored. Within each section, normal regions/stromal elements served as the internal control to assess quality of hybridization. A case was considered positive for translocation if >15% cells showed at least one fusion signal with or without a separate red and green signal representing the normal copy/allele. Amplification was defined as >10 copies or at least one small cluster of gene/locus (≥4 copies; resulting from tandem duplication/repeats). In cells with high-level amplification (HSR-type/tandem repeats or DM), signals beyond 20 could not be accurately counted and were therefore given a score of 20. Cells with 3–5 copies and 6–10 copies were considered to be polysomic and high polysomic, respectively.

**Cell culturing**. The HSG and TCG580 cell lines were obtained from O. Baker and G. Stenman, respectively. Short tandem repeat profiling was used to confirm cell line identity. HSG was generated as human salivary gland epithelium cell line and is commonly used in salivary gland research, but was later reported to be a derivative from HeLa cells (http://www.sigmaaldrich.com/europe/life-science-offers/cell-cycle/sigma-ecacc-cell-cancer-cell-lines.html). Our profiling showed that HSGs were genetically similar, but not identical, to HeLa cells. These cells were cultured in Dulbecco's Modified Eagle medium (DMEM) with 5% fetal bovine serum (FBS). The HEK-293 cell line (ATCC, Manassas, VA, USA) was cultured in DMEM with 10% FBS. Primary pleomorphic adenoma cells were obtained from G. Stenman and were cultured in DMEM with 10% FBS, 1% penicillin/streptomycin, 20 ng/ml epidermal growth factor, and 5 μg/ml insulin. All cell lines were regularly tested to exclude mycoplasma contamination.

**Lentiviral constructs**. For TGFBR3-PLAG1 experiments, we used lentiviral constructs expressing PLAG1 (GeneCopoeia, Rockville, MD, USA; catalog number EX-S0269-Lv105), TGFBR3 (EX-T0451-Lv204), or empty vectors EV1 (EX-NEG-Lv203, corresponding to the PLAG1 plasmid) and EV2 (EX-NEG-Lv203, corresponding to the TGFBR3 plasmid). For MSN-ALK experiments, wild-type MSN and MSN-ALK was amplified using cDNA extracted from the MSN-ALK-positive tumor, and cloned into the pHIV-Luc-ZsGreen vector (plasmid #39196, Addgene, Cambridge, MA, USA).

**Plasmid transfection**. Lentiviral constructs were transfected into HEK 293T cells using the expression vectors pCMV-dR8.2 (gag/pol) and VSV-G (env) in combination with Fugene 6 (Promega, Madison, WI, USA). Viral stocks were collected 48 h after transfection, filtered (0.45 μm) and placed on target cells for 8–12 h in the presence of 8 μg/ml polybrene. Infected cells expressing GFP and/or Cherry red were selected using a FACS aria flow cytometer (Becton Dickinson).

**Western blotting**. Cells were lysed in CelLytic M (Sigma-Aldrich) supplemented with Halt protease and phosphatase inhibitor cocktails (Thermo Scientific). Protein lysates were mixed with NuPAGE LDS sample buffer (Thermo Scientific), run in NuPAGE 4–12% Bis-Tris Gels (Thermo Scientific), and transferred to PVDF membranes for immunoblotting. The following antibodies were used: HA-tag (Cell Signaling Technology, Danvers, MA, USA; cat. no. 3724; rabbit monoclonal, 1:10,000), phospho-MEK1/2 (Cell Signaling, #9121; rabbit polyclonal, 1:800), phospho-ERK1/2 (Cell Signaling, #9101; rabbit polyclonal, 1:1000), ERK1/2 (Cell Signaling, #9102; rabbit polyclonal, 1:1000), phospho-STAT3 (Cell Signaling, #4113; mouse monoclonal, 1:800), and STAT3 (Cell Signaling, #9139; mouse monoclonal, 1:800). See Supplementary Fig. 35 for full pictures of blots that were cropped in main figures.

**Proliferation experiments**. For growth curve assays, 20,000 cells were seeded in triplicate in a 12-well plate and counted at different time points using the Vi-Cell XR Cell Viability Analyzer (Beckman Coulter, Brea, CA, USA).

**Migration experiments**. For migration analysis, 30,000 cells were seeded in each well on an xCELLigence CIM-plate 16. Migration index was analyzed hourly for 24 h using an xCELLigence RTCA DP instrument (ACEA Biosciences, San Diego, CA, USA).

**Soft agar colony formation assay**. A bottom layer of agar was created by adding 1.5 ml of a 1:1 mix of melted noble agar (1%) and 2× concentrated DMEM with 10% FBS in six-well plates. After incubation for 30 min at room temperature, 5000 cells were seeded in triplicates in a 1:1 mix of melted noble agar (0.6%) and 2× concentrated DMEM with 10% FBS. Plates were incubated at 37 °C for 3 weeks, and colonies were quantified with a GelCount™ colony counter (Oxford Optronix, Abingdon, UK). For ALK inhibitor experiments, crizotinib or lorlatinib was

added to the 2× concentrated media at double the final concentration before mixing it 1:1 with 0.6% noble agar.

**Drug sensitivity assay.** For ALK inhibitor sensitivity experiments, 20,000 cells were seeded in triplicates on a 12-well plate. After 24 h, cells were treated with indicated doses of crizotinib (Sigma-Aldrich), lorlatinib (MedChem Express, Monmouth Junction, NJ, USA), or DMSO (Sigma-Aldrich) for 72 h and then quantified using the Vi-Cell XR Cell Viability Analyzer (Beckman Coulter).

**Data availability.** All whole-exome and RNA sequencing bam files are available through the NIH Sequence Read Archive database (https://www.ncbi.nlm.nih.gov/sra/; accession number no. SUB2758934). All other relevant data are available from the authors.

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

## Acknowledgements

This work was supported by NIH P30 CA008748. M.G.D. was supported by Sahlgrenska University Hospital, The Swedish Society of Medicine, The Gothenburg Medical Society, and Svensson's Fund for Medical Research. L.G.T.M. was supported by the Damon Runyon Cancer Research Foundation, NIH K08 DE024774, Cycle for Survival, and the Jayme and Peter Flowers Fund. T.A.C. was supported by grants from the Adenoid Cystic Carcinoma Research Foundation. T.A.C. and A.L.H. were supported by the Geoffrey Beene Cancer Center. G.S. was supported by the Swedish Cancer Society and BioCARE—a Strategic Research Program in Cancer at University of Gothenburg. FISH analyses were co-funded by MSKCC Molecular Cytogenetics Core and NIH Cancer Center support grant P30 CA008748. We thank T. Nielsen for graphic design.

## Author contributions

M.G.D., N.K., T.A.C., and L.G.T.M. designed the study; M.G.D., M.P., K.W.L., Z.N., D.R., J.J.H., and K.C. performed experiments; N.K. and R.G. performed pathology reviews; M.G.D., N.K., M.P., K.W.L., V.M., A.D., L.A.W., L.W., F.K., G.J.N., I.G., N.R., G.S., and L.G.T.M. collected and analyzed data; M.G.D. and L.G.T.M. wrote the manuscript; N.K., M.P., K.W.L., A.D., A.L.H., C.R.A., G.S., and T.A.C. revised the manuscript and provided conceptual advice. T.A.C. and L.G.T.M. supervised the study.
