## [Peer Review File · Nature Communications]

Reviewers' comments:

Reviewer #1 (Remarks to the Author):

General.

This is a thorough look at myoepithelial carcinoma of the salivary glands (MECA), a tumour not all that well studied previously and in particular, this study performs the function of being a pretty comprehensive look at molecular alterations, particularly PLAG1 fusions.

The one question I have concerns the criteria for diagnosing MECA ex PA. Was this just on morphological grounds? One interesting paper looking at salivary duct carcinoma (SDC) ex PA found PLAG1/HMGA2 alterations in a few cases and postulated (I believe correctly) that the carcinoma overgrew and obscured the host PA. (Bahrami A, Perez-Ordoñez B, Dalton JD, Weinreb I. An analysis of PLAG1 and HMGA2 rearrangement in salivary duct carcinoma and examination of the role of precursor lesions. *Histopathology* 2013; 63: 250-262.). I wonder if a certain percentage of the apparently de novo MECAs in the present study had similarly arisen in PAs and overgrown their host adenomas. I think the authors answer this partially, but please could they make this more clear.

A minor point I note (and support) is that the authors correctly quote the 3rd edition of the WHO Blue Book, as the incidence information is not included in the most recent 4th edition.

References.

As far as I can see, these are generally up to date and correct. The exception is the 2013 Bahrami paper (see above).

Illustrations.

Generally clear and informative.

Language.

The text generally reads very well.

Reviewer #2 (Remarks to the Author):

The authors are to be commended on a very comprehensive study looking at the genetic landscape of salivary gland myoepithelial carcinoma. The sequencing identified several novel gene rearrangements, and the authors then pursued the nature of these novel findings. Their initial mechanistic work is intriguing and demonstrates the potentially oncogenic roles of these fusion genes.

The analysis for the identification of these fusions and the sequencing methodology appears sound. There was solid validation work done to verify the presence of these alterations using FISH and RNA seq as well in a subset of samples.

It is of note that the main negative prognostic factor described is increased copy number variation, which was demonstrated in the final analysis. However, there was no other associations described. I would like to see the relationship between basic cancer staging and the presence of the fusions and influence of staging on survival above and beyond the CNV alterations.

Otherwise, I was excited to read about this in depth analysis about an underserved tumor type, and even more interested to read about the presence of these unusual fusion products, which appears to be a hallmark of many other salivary gland malignancies.

Point-by-point response to reviewers' comments

Reviewer #1 (Remarks to the Author):

General.

This is a thorough look at myoepithelial carcinoma of the salivary glands (MECA), a tumour not all that well studied previously and in particular, this study performs the function of being a pretty comprehensive look at molecular alterations, particularly PLAG1 fusions.

Response:

We thank the reviewer for the positive comments.

The one question I have concerns the criteria for diagnosing MECA ex PA. Was this just on morphological grounds? One interesting paper looking at salivary duct carcinoma (SDC) ex PA found PLAG1/HMGA2 alterations in a few cases and postulated (I believe correctly) that the carcinoma overgrew and obscured the host PA. (Bahrami A, Perez-Ordoñez B, Dalton JD, Weinreb I. An analysis of PLAG1 and HMGA2 rearrangement in salivary duct carcinoma and examination of the role of precursor lesions. *Histopathology* 2013; 63: 250-262.). I wonder if a certain percentage of the apparently de novo MECAs in the present study had similarly arisen in PAs and overgrown their host adenomas. I think the authors answer this partially, but please could they make this more clear.

Response:

We agree. We have added a paragraph in the Discussion section (page 15) addressing this valid point, and included the suggested reference.

A minor point I note (and support) is that the authors correctly quote the 3rd edition of the WHO Blue Book, as the incidence information is not included in the most recent 4th edition.

References.

As far as I can see, these are generally up to date and correct. The exception is the 2013 Bahrami paper (see above).

Illustrations.

Generally clear and informative.

Language.

The text generally reads very well.

Response:

We thank the reviewer for these helpful suggestions.

Reviewer #2 (Remarks to the Author):

The authors are to be commended on a very comprehensive study looking at the genetic landscape of salivary gland myoepithelial carcinoma. The sequencing identified several novel gene rearrangements, and the authors then pursued the nature of these novel findings. Their initial mechanistic work is intriguing and demonstrates the potentially oncogenic roles of these fusion genes.

The analysis for the identification of these fusions and the sequencing methodology appears sound. There was solid validation work done to verify the presence of these alterations using FISH and RNA seq as well in a subset of samples.

Response:

We thank the reviewer for these supportive comments.

It is of note that the main negative prognostic factor described is increased copy number variation, which was demonstrated in the final analysis. However, there was no other associations described. I would like to see the relationship between basic cancer staging and the presence of the fusions and influence of staging on survival above and beyond the CNV alterations.

Response:

We have added more information in the Results section (pages 13-14), and included Supplementary Figures 31 and 32 as a response to this interesting notion. We found an association between sex and recurrence (Supplementary Fig. 31), but staging and fusion status were not associated with clinical outcomes. Since we found no significant correlation between fusion status and disease stage at diagnosis, we chose to not include that information as a figure, but we added a notion in the text. We also show that the CNAs are associated with recurrence, within each stage category (Supplementary Fig. 32).

Otherwise, I was excited to read about this in depth analysis about an underserved tumor type, and even more interested to read about the presence of these unusual fusion products, which appears to be a hallmark of many other salivary gland malignancies.

Response:

Thank you for your helpful suggestions.

We have also added Supplementary Fig. 33, with full pictures of gels and blots that have been cropped for inclusion in main figures.

REVIEWERS' COMMENTS:

Reviewer #1 (Remarks to the Author):

As I stated in my original review, this is a thorough look at molecular alterations in myoepithelial carcinoma of the salivary glands (MECA), particularly PLAG1 fusions. The one question I had concerning the criteria for diagnosing MECA ex PA, has been answered.

Reviewer #2 (Remarks to the Author):

The authors have been responsive in their edits and have adequately addressed the reviewers' suggestions.

Point-by-point response to reviewers' comments

Reviewer #1 (Remarks to the Author):

As I stated in my original review, this is a thorough look at molecular alterations in myoepithelial carcinoma of the salivary glands (MECA), particularly PLAG1 fusions. The one question I had concerning the criteria for diagnosing MECA ex PA, has been answered.

Response: We thank the reviewer for the positive feedback.

Reviewer #2 (Remarks to the Author):

The authors have been responsive in their edits and have adequately addressed the reviewers' suggestions.

Response: Thank you.